# Emodin Sensitizes Cervical Cancer Cells to Vinblastine by Inducing Apoptosis and Mitotic Death

**DOI:** 10.3390/ijms23158510

**Published:** 2022-07-31

**Authors:** Wojciech Trybus, Ewa Trybus, Teodora Król

**Affiliations:** Department of Medical Biology, Institute of Biology, The Jan Kochanowski University, Uniwersytecka 7, 25-406 Kielce, Poland; ewa.trybus@ujk.edu.pl

**Keywords:** emodin, vinblastine, mitotic death, apoptosis, oxidative stress

## Abstract

In recent years, studies on the effects of combining novel plant compounds with cytostatics used in cancer therapy have received considerable attention. Since emodin sensitizes tumor cells to chemotherapeutics, we evaluated changes in cervical cancer cells after its combination with the antimitotic drug vinblastine. Cellular changes were demonstrated using optical, fluorescence, confocal and electron microscopy. Cell viability was assessed by MTT assay. The level of apoptosis, caspase 3/7, Bcl-2 protein, ROS, mitochondrial membrane depolarization, cell cycle and degree of DNA damage were analyzed by flow cytometry. The microscopic image showed indicators characteristic for emodin- and vinblastine-induced mitotic catastrophe, i.e., multinucleated cells, giant cells, cells with micronuclei, and abnormal mitotic figures. These compounds also increased blocking of cells in the G2/M phase, and the generated ROS induced swelling and mitochondrial damage. This translated into the growth of apoptotic cells with active caspase 3/7 and inactivation of Bcl-2 protein and active ATM kinase. Emodin potentiated the cytotoxic effect of vinblastine, increasing oxidative stress, mitotic catastrophe and apoptosis. Preliminary studies show that the combined action of both compounds, may constitute an interesting form of anticancer therapy.

## 1. Introduction

Cervical cancer is one of the leading causes of death among women and is a major problem, especially in developing countries. The main cause of the incidence of this type of cancer is persistent human papillomavirus (HPV) infections, of which HPV 16 and 18 are the most carcinogenic types. Genetic and epigenetic factors also contribute to cancer transformation [1]. The high mortality rate is related to the difficulty of its detection in the early stages, which is related to the lack of specific symptoms associated with the pre-invasive stage, and in most cases the disease appears to be diagnosed when already in advanced stages [2]. Currently, cervical cancer treatment strategies include screening techniques as well as surgery, radiation therapy, hormone therapy, and chemotherapy [3]. However, these methods are not always successful, because the main reasons for failure are either chemoresistance or radio resistance [4].

Since drug resistance is a significant obstacle to the successful treatment of cancer, it is important to use a combination therapy that combines two or more anti-cancer drugs. Their combination increases the effectiveness of therapy because it influences, among other things, the reduction in tumor growth, the cessation of cell division, or the induction of apoptosis [5]. An example of such combined action of two different compounds is vinblastine and emodin, as presented in this work.

Vinblastine is one of more than 130 indole alkaloids found in *Catharanthus roseus*; it exhibits potent pharmacological activity and is used in chemotherapy [6]. Indole alkaloids exhibit potent anticancer effects through various antiproliferative mechanisms [7]. In recent years, research has shown that indole alkaloids are involved in the regulation of autophagy, which may contribute to increasing their effectiveness in the prevention and treatment of cancer. This is due to the influence of alkaloids on various signaling pathways PI3K/Akt/mTOR, MAPK, ROS activity, and the level of Beclin 1 [8]. In studies conducted on various cell lines, vinblastine has been shown to induce apoptosis [9,10]. In recent years, it has also been shown that alkaloids have great therapeutic potential as a result of their combined action with other anticancer drugs, including with doxorubicin, prednisone, cyclophosphamide, rituximab, dacarbazine, or bleomycin in the treatment of leukemia, malignant granulomatosis (Hodgkin’s disease), non-Hodgkin’s lymphoma, chronic lymphocytic leukemia, and testicular, breast, bladder, and lung cancer [11]. Vinblastine is a specific alkaloid whose influence on the dynamics of the cytoskeleton depends on the concentration used. At low nanomolar concentrations (0.1–3 nM), vinblastine suppresses the dynamic instability preferentially at the plus microtubule ends without changing the polymer mass of the microtubules. Intermediate drug concentrations (30 nM to 2 µM) depolymerize microtubules quickly, while at very high concentrations (≥10 µM), it inhibits microtubule polymerization and induces tubulin aggregation, including the formation of paracrystalline structures. Thus, the consequence of inhibiting tubulin polymerization is that vinblastine induces an intense mitotic catastrophe. The induction of a mitotic catastrophe by disruption of microtubules is an established goal of anti-cancer therapy, and the ensuing apoptotic or non-apoptotic cell death remains poorly understood [12].

Anthraquinones are bioactive natural products, and some of them are ingredients in drugs mainly used in Chinese medicine [13]. These compounds exert multidirectional effects on the body, including: laxative [14], anti-inflammatory [15], antibacterial [16], antifungal, and antiviral, including inhibition of the SARS-CoV-2 coronavirus [17]. They also exhibit immunomodulatory [18], anti-cancer [19], and neuroprotective effects [20]. One of the anthraquinones with high biological activity is emodin, which is most often extracted from *Aloe barbadensis* or from plants from the *Polygonaceae* family, such as *Rheum palmatum* and *Rheum officinale*, which are used in traditional Chinese medicine. Emodin shows, inter alia, anti-inflammatory [21,22], antioxidant [23], anticancer [24,25], and anti-angiogenic properties, which have been studied both at the in vitro and in vivo level [26,27]. It has also been shown that this compound sensitizes cancer cells to chemotherapeutic agents, which may be important in oncological therapy [28,29,30,31,32]. By studying the metabolism of anthraquinones, it has been shown that their conversion to another anthraquinone can increase the concentration of the latter in the blood, which simultaneously enhances pharmacological and/or toxicological effects [13]. Hence, the great interest in these compounds is the basis for making modifications in their structure, contributing to the development of new compounds that may be promising anticancer agents in the future [33]. 

As shown in the literature, emodin is an anthraquinone that exhibits selective toxicity to various cancer cells, and the range of emodin concentrations used (most often 1–100 µM), as well as the duration of action, varies (24–72 h). Often-tested concentrations of emodin, as in our work, include 40 and 80 µM. The concentrations used confirm that this compound may have the effect of increasing the sensitivity of cancer cells to the action of vinblastine without the use of highly toxic doses, which is very important in anti-cancer therapy. In contrast to the apoptosis of cancer cells, which requires the use of high concentrations of chemotherapeutic agents, research results show that mitotic catastrophe is activated in response to significantly lower doses of therapeutic agents [34].

Therefore, in the context of our study, it seemed interesting to combine the well-known cytostatic-vinblastine with emodin, which, according to literature reports, sensitizes human cancer cells to chemotherapeutic agents [28,35]. 

## 2. Results

### 2.1. Emodin and Vinblastine Decrease Cell Viability

The demonstrated increase in the number of apoptotic cells was correlated with the gradual increase in dead cells shown in the MTT test (Figure 1A). At 40 µM and 80 µM concentration of emodin, a decrease in cell viability was observed, down to 66.09% (*p* ≤ 0.0001) and 35.13% (*p* ≤ 0.0001), respectively. Additionally, vinblastine inhibited the viability of the tested cells (36.07% viable cells, *p* ≤ 0.0001). An increased percentage of dead cells was observed as a result of the combined action of emodin and vinblastine, to 71.51% (E 40 µM + VBL 10 µM) and to 80.36% (E 80 µM + VBL 10 µM), at *p* ≤ 0.0001.

### 2.2. Emodin and Vinblastine Induce Cell Apoptosis by Activating Caspase 3/7

The 24 h incubation of cells with emodin, depending on the concentration, resulted in a gradual growth of apoptotic cells (test with annexin V). At 40 µM of emodin, apoptotic cells accounted for 47.8% (*p* ≤ 0.0001), and at 80 µM, the percentage of apoptotic cells increased to 58.7% (*p* ≤ 0.0001) (Figure 1B,C). Vinblastine at a concentration of 10 µM increased the percentage of apoptotic cells to 39.2% (*p* ≤ 0.0001) compared to control (3.96%). The consequence of exposure of HeLa cells to the combined action of emodin (40 µM and 80 µM) and vinblastine (10 µM) was the intensification of the process of apoptosis. There was a significant increase in the percentage of cells (74.98%) in late-stage apoptosis at 40 µM concentration of emodin, and 81.05% at 80 µM concentration, at *p* ≤ 0.0001. 

At 40 µM and 80 µM concentrations of emodin, caspase-positive cells accounted for 25.58% and 34.38%, respectively. After treatment with vinblastine (10 µM), the number of cells with active caspase 3/7 was 21.35%. The highest activity of the assayed caspase was found after the combined treatment of the cells with vinblastine and emodin, where at a concentration of 80 µM emodin and 10 µM vinblastine, over 87% (*p* ≤ 0.0001) were apoptotic cells (Figure 2A,B).

### 2.3. Emodin and Vinblastine Inactivate the Bcl-2 Protein

Both emodin and vinblastine induced inactivation of the anti-apoptotic protein Bcl-2 in HeLa cells (Figure 3A,B). At emodin concentrations of 40 µM and 80 µM, cells with Bcl-2 inactivated accounted for 49.8% and 59.3% (*p* ≤ 0.0001), respectively, compared to the control (4.3%). Vinblastine also caused inactivation of the above-mentioned protein to the extent of 48.4% (*p* ≤ 0.0001). Progressive changes were demonstrated with the use of emodin in combination with vinblastine of up to 83.9% (E 40 µM + VBL 10 µM) and of up to 87.8% (E 80 µM + VBL 10 µM), respectively, at *p* ≤ 0.0001. This is confirmed by the pro-apoptotic mechanism of the combined action of vinblastine and emodin through the mitochondrial pathway.

### 2.4. Emodin and Vinblastine Induce Mitotic Death

Analyzing the changes in the nuclei of the tested cells (DAPI staining) after the treatment with emodin, it was shown (depending on the concentration) that there was an increase in the number of cells with altered shape, apoptotic cells with condensed chromatin, multinucleated cells, including cells with micronuclei (Figure 4) and abnormal mitotic figures (three-polar anaphase). Vinblastine (10 µM) induced changes in the mitotic apparatus, as expressed by an increase in the number of cells with micronuclei, as well as multinucleated and giant cells with numerous micronuclei. The response to the action of the alkaloid used was the presence of cells with a disturbance in the formation of chromosomes, especially at the stage of prometaphase. The consequence of the combined action of emodin and vinblastine was the intensification of changes proving a mitotic catastrophe. Numerous cells with mitotic abnormalities and chromosomal disorganization were also shown, which corresponded to the presence of micronucleated cells, multinucleated cells, and giant cells, both mono- and multinucleated.

### 2.5. Emodin and Vinblastine Induce Phosphorylation of ATM Kinase

The 24 h effect of the tested anthraquinone, vinblastine, and the combination of their actions contributed to an increase in the number of cells with phosphorylation. An increase in ATM was demonstrated in 24.41% (*p* ≤ 0.0001) of the pool of analyzed cells for emodin (40 µM) and in 35.06%, *p* ≤ 0.0001 for emodin (80 µM) compared to 2.68% for the control. Vinblastine at a concentration of 10 µM increased the ATM activity to 38.58%. In contrast, the combined effect of emodin (40 and 80 µM) with vinblastine (10 µM) further increased the number of cells with kinase phosphorylation, to 44.78% and 55.11%, respectively, at *p* ≤ 0.0001 (Figure 5A,B). However, no histone phosphorylation of γH2AX or an increase in DSB-type lesions was demonstrated. The observed increase in phosphorylation of ATM kinases may indicate the activation of oxidative stress induced by the tested compounds.

### 2.6. Emodin and Vinblastine Induced Ultrastructural Changes in Cervical Cancer Cells

Submicroscopic changes in HeLa cells are shown in Figure 6A. After a 24 h treatment with 40 µM emodin, an increased number of Golgi apparatuses with swollen cisterns was observed. The 80 µM concentration of emodin activated the lysosomal system represented by primary lysosomes and autophagic vacuoles. The cytoplasm also contained swollen mitochondria and numerous swollen Golgi apparatuses with dispersed cisterns. On the basis of morphometric measurements, a swelling to 0.78 µm with 40 µM and to 0.89 µm with 80 µM concentrations of emodin was shown (Figure 6B). Vinblastine (10 µM) caused the swelling of high-amplitude mitochondria to 1.72 µm. Mitochondria were characterized by a translucent matrix and shortened cristae drawn into the membrane. Numerous elements of the cytoskeleton, especially microtubules, were also present. The consequence of the combined action on HeLa cells of emodin (40 µM and 80 µM) and vinblastine (10 µM) was a significant swelling of the mitochondria (over 2 µm). In addition, the mitochondria had a completely translucent matrix and damaged mitochondrial cristae.

### 2.7. Emodin and Vinblastine Induce ROS Production and Changes in Mitochondria in HeLa Cells

It follows from Figure 7 that treatment of HeLa cells with emodin and vinblastine results in a concentration-dependent and co-action-dependent increase in reactive oxygen species (ROS) (Figure 7A,B). Compared to the negative control (no exposure), where ROS (+) cells accounted for 4.59% of the pool of cells exposed to 40 µM emodin, ROS (+) cells accounted for 23.3%, and this percentage increased to 38.5% at the highest concentration (80 µM). Additionally, vinblastine at a concentration of 10 µM induced an increase in ROS levels to 37.51%. On the other hand, a significant increase in ROS in cells was observed during the combined action of emodin and vinblastine. When emodin at concentrations of 40 µM and 80 µM was combined with vinblastine (10 µM), an increased generation of reactive oxygen species to 52.3% (*p* ≤ 0.0001) and 55.72% (*p* ≤ 0.0001), respectively, was demonstrated. Exposing HeLa cells to the separate and combined action of emodin and vinblastine intensified the generation of ROS (especially with the combined action of both compounds) and contributed to the induction of oxidative stress, which may lead to cell death. At the same time, a decrease in the mitochondrial membrane potential was found (Figure 7C,D). The consequence of the combined action of emodin (80 µM) and vinblastine (10 µM) was an increase in the number of dead cells with a damaged membrane to over 90% (*p* ≤ 0.0001), which was correlated with a significant increase in the number of cells in late apoptosis (Figure 1B) and with observed microscopic changes (Figure 7E). It was shown that with the increase in the concentration of the tested compounds, a gradual extinction of the green fluorescence emission from mitochondria labeled with rhodamine 123 was visible. Cells treated with vinblastine, combined with emodin, were also characterized by altered morphology; numerous shrunken and rounded cells were visible, and these changes indicated ongoing apoptosis.

### 2.8. Emodin and Vinblastine Induce Morphological Changes in HeLa Cells

As a result of 24 h exposure of cells to the action of emodin, the number of cells undergoing a mitotic catastrophe increased. A significant increase in the number of binuclear cells, multinucleated cells with micronuclei was observed (Figure 8A). There was also an increase in the number of giant cells, which were characterized by significantly enlarged sizes and the presence of large different-shaped nuclei.

Vinblastine caused changes like emodin, i.e., an increased percentage of multinucleated cells (19.8%, number of cells = 594, *p* ≤ 0.0001) with nuclei different in terms of both size and shape, and an increased percentage of cells with micronuclei (23.8%, number of cells = 714, *p* ≤ 0.0001) present in the cytoplasm, which may reflect abnormal chromosome segregation (Figure 8C). An increase in the percentage of giant cells was also demonstrated, and the analyzed changes were statistically significant (*p* ≤ 0.0001). Characteristic for the tested vinblastine concentration (10 µM) was also an increase in the percentage of cells with a chromosome formation disorder, which was most likely an effect of cytoskeleton depolymerization. The combined action of vinblastine (10 µM) with emodin at a concentration of 40 µM intensified the mitotic catastrophe, which was reflected in a statistically significant more than 21-fold increase in the number of binuclear cells (19.13%, *p* ≤ 0.0001), a 99-fold increase in the number of multinuclear cells (21.16%, *p* ≤ 0.0001), and a 238-fold increase in the number of cells with micronuclei (23.23%, *p* ≤ 0.0001) in relation to cells in the control group. The consequence of the combined action of vinblastine (10 µM) and emodin (80 µM) was also the presence of cells at the prometaphase stage, with dispersed chromosomes, which constituted 21.6% (*p* ≤ 0.0001) of all analyzed cells.

### 2.9. Emodin and Vinblastine Disrupt the Cell Cycle

Cytometric analysis showed that emodin and vinblastine induced marked cell arrest in the G2/M phase characteristic of cytostatics. As the concentration of emodin increased, an increasing percentage of cells blocked in the G2/M phase was observed, i.e., 63.54% (*p* ≤ 0.0001) at a concentration of 40 μM and 74.27% (*p* ≤ 0.0001) at a concentration of 80 μM, compared to controls (36.2%) (Figure 8E,F). Similar changes in the cell cycle were shown by vinblastine (10 µM), where 82.61% (*p* ≤ 0.0001) were cells blocked in the above-mentioned phase. The combined action of emodin and vinblastine resulted in an increased accumulation of cells blocked in the G2/M phase, dependent on the concentration of emodin. These values were 87.33% (*p* ≤ 0.0001) at the lower concentration of anthraquinone (40 µM E + 10 µM VBL) and 91% (*p* ≤ 0.0001) at the higher concentration (E 80 µM + 10 µM VBL).

### 2.10. Emodin and Vinblastine Reorganize the Cytoskeleton

As a result of the action of emodin, changes in the structure of the cytoskeleton of the studied cells were observed. Emodin at a concentration of 40 μM caused a slight reorganization of the cytoskeleton and a change in the shape of the cells. Small clusters of actin were also visible in the cytoplasm (Figure 9). On the other hand, emodin at a concentration of 80 μM induced changes in the cytoskeleton indicating its damage, which was expressed in a reduction in the number of or absence of actin fibers, as well as the presence in the cytoplasm of cells of large actin clusters, often distributed around the nucleus. The response to the toxic effects of vinblastine was an increase in the number of rounded cells with damaged spindle and cells in prometaphase.

The combined effect of vinblastine and emodin was the presence of cells with visible actin deposits and rounded cells with condensed actin. Giant, multinucleated cells with disrupted distribution of actin microfilaments were visible, characteristic of the mitotic catastrophe. Also present were shrunken apoptotic cells with condensed nuclear chromatin, with the arrangement of the actin cytoskeleton around the nucleus characteristic of this type of cell death.

## 3. Discussion

Antimitotic drugs are among the most effective chemotherapeutic compounds currently used in the treatment of cancer. These compounds, by binding to microtubules, inhibit the function of the mitotic spindle, and their effect is expressed, inter alia, in the form of cell cycle arrest and induction of apoptosis in cancer cells [36]. An example of such a drug is vinblastine, which binds to tubulin and blocks the formation of microtubules, thus inhibiting cell division [37]. It also shows a pro-apoptotic effect [9,38,39] and induces changes in the lysosomal compartment [40,41].

Emodin is an anthraquinone that has a strong cytotoxic effect on cancer cells of the lung, liver, colon, breast, pancreas, cervix, and ovaries, and in leukemia [42], and its anti-cancer properties are associated with a pro-apoptotic effect [43], anti-angiogenic [31], or with induction of lysosomal cell death [25]. Our research also shows that this compound, influencing the cytoskeleton, induces a mitotic catastrophe [24].

Because the importance of combination therapies is increasingly being emphasized, relying on the combination of different anticancer drugs to increase the effectiveness of oncological treatment, the combined action of vinblastine and emodin significant importance may have.

Compared to monotherapy, combination therapy may reduce drug resistance and produce positive effects, which may be expressed in terms of, i.e., a reduction in tumor growth and metastasis potential, blockage of cell division, reduction in the population of cancer stem cells, and the induction of various types of cell death [5].

Mitotic catastrophe is one of the types of cell death, and is the aftermath of cell cycle disorders, abnormal chromosome segregation, and cell divisions. Its morphological expression is the presence of cells with micronuclei, multinucleated cells, cells with multipolar mitosis, and giant cells [44,45,46]. All these mitotic markers were demonstrated in our study by the action of both emodin and vinblastine on cervical cancer cells (Figure 8). 

Vinblastine at the tested concentration (10 µM) clearly inhibited microtubule polymerization, resulting in an intense mitotic catastrophe and the presence of numerous micronucleations. However, it should be emphasized that the antimitotic effect of vinblastine was clearly enhanced by emodin (mainly 40 µM), as evidenced by the increased number of cells with abnormal segregation of chromosomes in the cytoplasm. These compounds probably act synergistically by damaging the mitotic spindle, which affects the disorganization of the cytoskeleton in the tested cells, consequently leading to the inhibition of cell division.

The mechanism of action of vinblastine, as well as other antimitotic drugs, is primarily related to the inhibition of cell division [47]. During mitotic cell division, the formation of microtubules—fibrous protein structures made of smaller subunits, i.e., tubulin protein molecules—is essential [48]. Microtubules are dynamic structures that undergo constant remodeling with rapidly changing lengths, and vinblastine prevents their reorganization and the formation of a dividing spindle, which stops the mitosis process at a particular stage. This is related to the disturbance of vital functions of the cell, which results in its death. The action of vinblastine applies particularly to rapidly dividing cells, such as cancer cells; therefore, by inhibiting cell division, their growth and, consequently, the development of cancer, is limited [49,50].

Furthermore, emodin is an anthraquinone that damages the division spindle, as expressed by the presence of numerous and dispersed cisternae in the Golgi apparatuses in the ultrastructure of the examined cells, proving their disorganization (Figure 6). The role of the cytoskeleton in the organization of the Golgi apparatus has been confirmed by literature data [51,52]. Emodin also induced the degradation of phalloidin-labeled actin filaments, as well as the formation of its aggregates in the cytoplasm of cells. Similar changes have also been observed in the tested cells following the action of another anthraquinone, aloe-emodin [53]. The effect of the combined action of emodin and vinblastine was the intensification of changes in the structure of the cytoskeleton, as documented in Figure 9, where numerous actin deposits can be observed, and its condensation is also visible in the rounded cells. 

The mitotic catastrophe was also confirmed by changes in the cell cycle, i.e., cell arrest mainly in the G2/M phase of the cycle, both after treatment with emodin and vinblastine (Figure 8). These changes are characteristic of most cytostatics used in oncological treatment.

A mitotic catastrophe is characterized by the absence or delay of cells entering the G1/S cell cycle checkpoint, or by cell arrest in the G2 phase. Neoplastic cells usually have inactive checkpoints on the border of G1/S phases, which, after the action of DNA damaging factors, leads to the cycles stopping in the G1 phase. As a consequence of this, there is a transient accumulation of cells in the G2/M phase, which leads to the entry of cells into the path of mitosis, and which is expressed by mitotic death [37,45,46,54,55].

The mitotic catastrophe has also been observed in other cancer cells after the action of bortezomide [56], etoposide, cisplatin, staurosporin, and taxol [57], which are chemotherapeutic agents used in oncology. Regardless of the mechanism of action, these compounds induce defects in the mitotic apparatus, which are expressed in the form of microtubule destabilization, lack of chromosome segregation, abnormal divisions, and consequently induction of a mitotic catastrophe [46]. As a result of the combined action of 80 µM emodin and vinblastine, a gradual reduction in the indicators of mitotic catastrophe was observed (binuclear cells, multinucleated cells, or cells with micronuclei). The combined action of the tested compounds increased the cytotoxicity to HeLa cells, and the consequence of this was an increase in the percentage of apoptotic cells, including indicators of mitotic catastrophe. According to the literature, the last stage of mitotic death is the direction of cells to apoptotic path, which is associated with the increased permeability of mitochondrial membranes and the activation of caspases [54,55]. Hence, the number of studied mitotic indicators decreases at the expense of an increase in the number of apoptotic cells. Additionally, cytometric analysis showed that emodin and vinblastine, as well as these compounds added together, had a pro-apoptotic effect when added to the cells, as evidenced by a significant increase in apoptotic cells with active executive caspase 3 and 7, and an increase the number of cells with Bcl-2 inactivation. The pro-apoptotic effect of the tested compounds was dependent on the generation of reactive oxygen species, which induced oxidative stress in HeLa cells, and the consequence of this was the activation of programmed cell death. 

Revealed as being a result of the combined action of emodin and vinblastine, the changes in the shape of the testes and the abnormal mitotic figures may be related to the ability of the tested compounds to interact with nuclear DNA. Our research shows that these compounds did not induce double-strand breaks (DSBs) in chromosomes, but induced ATM kinase phosphorylation, the greatest increase of which was shown after the combined action of emodin and vinblastine.

Additionally, Geriyol et al. [58] and Mhaidat et al. [59] demonstrated the genotoxic effect of *Vinca* alkaloids without double-stranded DNA breaks. The literature shows that other anthraquinones also exhibit genotoxicity, e.g., chrysophanol [60], physcion [61], aloe-emodin [62,63], danthron [64], and emodin, which is the subject of our research, and which, thanks to its structure, can easily penetrate between the adenine and thymine of the DNA double helix [65]. ATM kinase plays a key role in the response to DNA damage; however, attention is currently being paid to its important role as a modulator of oxidation (in response to stress), and as a modulator of mitochondrial homeostasis and apoptosis [66]. Therefore, the activation of ATM kinase observed in our research without damaging the DSB should be related to the simultaneous induction of oxidative stress. The highest concentration of the analyzed kinase, which was demonstrated after the combined action of emodin and vinblastine, could be a consequence of the high level of reactive oxygen species generated by the tested compounds (Figure 7). According to the research of Kamsler et al. [67], Guo et al. [68], Navrkalova et al. [69], and Ambrose et al. [70], an increase in reactive oxygen species contributed to ATM activation without a concomitant increase in DSBs. This can also be applied to the changes in the ultrastructure of the mitochondria, which are intracellular sources of reactive oxygen species, as shown in our research. They were characterized by altered structural organization, significant swelling, often including damage to the cristae, as well as a reduction in the mitochondrial membrane potential. The demonstrated increase in ROS concentration affects the intracellular pathways of survival or death, depending on the intensity of stimuli acting on the cell [66].

According to the conducted studies, emodin may enhance the cytotoxic effect of vinblastine in cervical cancer cells. Furthermore, in the work of Li [28], it was shown that emodin increased the cytotoxicity of another plant compound used in chemotherapy, which is paclitaxel. Emodin sensitized human ovarian cancer cells to the action of taxol, which was associated with the reduction of P-glycoprotein, XIAP, survivin levels and the induction of apoptosis. Taxol blocks the proliferation of rapidly dividing cells by interacting with the mitotic spindle [71,72], tightly binds to microtubules, prevents the release of tubulin subunits, and allows microtubules to grow but not shorten [73]. It should be emphasized that, despite the differences in the effects of taxol and vinblastine at the molecular level, taxol ultimately exerts the same cellular effects as vinblastine, i.e., it stops cells dividing during the stage of mitosis.

## 4. Materials and Methods

### 4.1. In Vitro Culture Conditions

HeLa (cervical carcinoma) cells were purchased from the American Type Tissue Culture Collection (Rockville, MD, USA). Cells were grown in DMEM (GIBCO, New York, NY, USA) with 10% fetal bovine serum (Biowest, Nuaillé, France) and an antibiotic mixture containing amphotericin B, penicillin G and streptomycin (Corning, Manassas, VA, USA) in a CO_2_ DirectHeat incubator (Thermo Fisher Scientific, Waltham, MA, USA). Emodin (C_15_H_10_O_5_) and vinblastine (C_46_H_58_N_4_O_9_) were purchased from Sigma-Aldrich (St. Louis, MO, USA). Cells were incubated with emodin in the concentration of 40 µM and 80 µM, with vinblastine in the concentration of 10 µM, and with emodin (40 µM and 80 µM) in combination with vinblastine (10 µM).

### 4.2. Assessment of Cell Viability

The principle of the test is based on the ability to convert the yellow MTT dye to purple formazan crystals by means of mitochondrial dehydrogenase in living cells. After 24 h of incubation with the test agents, cells (1 × 10^5^/mL) were incubated for 2 h with a solution of MTT (1 mg/mL) (3-(4,5-dimethylthiazol-2-yl)-2-5-diphenyltetrazolium bromide) (Sigma Aldrich, St. Louis, MO, USA) in 96-well plates (Falcon Fisher Scientific, Waltham, MA, USA). The resulting formazan crystals were then dissolved in DMSO (Sigma Aldrich, St. Louis, MO, USA) and the absorbance was measured at 570 nm using a Synergy 2 multi-detected microplate reader (BioTek, Winooski, VT, USA). The experiment was repeated 3 times.

### 4.3. Detection of Apoptosis

Apoptosis was assessed using Annexin V Dead Cell Kit (Merck KGaA, Darmstadt, Germany). After 24 h incubation with test compounds, cells at a density of 3 × 10^5^/mL were trypsinized by means of 0.25% trypsin-EDTA solution (Corning, Manassas, VA, USA) and centrifuged. Cells were then stained for 20 min in the dark, at room temperature, by adding 100 µL of annexin V-PE/7-AAD. The fluorescence intensity was analyzed on a Muse analyzer (Merck-Millipore, Burlington, MA, USA). The experiment was repeated 3 times.

### 4.4. Caspase 3/7 Activity Test

Caspase activity level was measured using the Muse Caspase-3/7 Kit (Merck-Millipore, Guyancourt, France). After 24 h of incubation with vinblastine and emodin, cells at a density of 3 × 10^5^/mL were trypsinized, centrifuged to pellet receive, and then incubated for 30 min at 37 °C with 5 µL Caspase-3/7 working solution according to protocol. The number of caspase-positive cells was determined using a Muse^®^ analyzer (Merck-Millipore, Guyancourt, France). The experiment was repeated 3 times.

### 4.5. Assessment of Bcl-2 Protein Phosphorylation 

Changes in Bcl-2 phosphorylation in HeLa cells (density of cells = 3 × 10^5^/mL) were assessed using the Muse™Bcl-2 Activation Dual Detection Assay Kit (Merck-Millipore, Guyancourt, France) according to the manufacturer’s instructions. Two direct conjugated antibodies were used in the kit, i.e., phospho-specific anti-phospho-Bcl-2 (Ser70)-Alexa Fluor^®^ 555 and a conjugated anti-Bcl-2-PECy5 antibody to measure total Bcl-2 expression levels. Evaluation of the degree of activation of the Bcl-2 pathway was performed by measuring the phosphorylation of Bcl-2 relative to the total expression of Bcl-2 in the tested cells. The experiment was repeated 3 times.

### 4.6. Measurement of the Reactive Oxygen Species Generation 

Cells (3 × 10^5^/mL) treated with vinblastine and emodin were assessed by determining the percentage of cells exposed to oxidative stress. For this purpose, after 24 h of exposure to the test compounds, the cells were stained with the Muse Oxidative Stress Kit (Merck Millipore, Guyancourt, France). Cells were treated with Muse Oxidative Stress Reagent working solution (190 µL), then both control and test samples were incubated for 30 min at 37 °C and analyzed to determine the percentage of ROS (−) negative cells and ROS cells (+) with active ROS. The experiment was repeated 3 times.

### 4.7. Measurement of the Mitochondrial Membrane Potential (Δψm)

The reduction of Δψm was analyzed with the Muse Mitopotential Assay kit (Merck Millipore, Guyancourt, France). Cells at a density of 3 × 10^5^/mL after treatment with vinblastine and emodin were suspended in the MitoPotential working solution and incubated at 37 °C for 20 min. After incubation, cells were stained by adding 5 µL of 7-AAD (dead cell marker) at room temperature for 5 min. The cell suspension was then analyzed by flow cytometry. The experiment was repeated 3 times.

### 4.8. Microscopic Evaluation of Changes in Mitochondrial Membrane Potential

After 24 h exposure to emodin and vinblastine, cells (1 × 10^5^/mL) were fixed in 3.7% paraformaldehyde and then incubated for 30 min with rhodamine 123 (Sigma Aldrich, St. Louis, MO, USA) at 5 µg/mL ethanol, a fluorochrome that binds to metabolically active mitochondria. Cells were then washed with PBS and analyzed under a Nikon A1R confocal microscope based on a Nikon Eclipse T*i* inverted microscope (Nikon Instruments Inc., Melville, NY, USA) and equipped with Nikon Nis Elements AR software (Nikon Instruments Inc., Melville, NY, USA). The experiment was repeated 3 times.

### 4.9. DAPI Staining

Evaluation of changes in cell nucleus morphology was performed using 4’,6-diamidino-2-phenylindole (DAPI) staining. After 24 h incubation of cells (1 × 10^5^/mL) with emodin, vinblastine, and emodin and vinblastine, DAPI (2.5 µg/mL) staining (Sigma Aldrich, St. Louis, MO, USA) was performed. For this purpose, cells were fixed in 3.7% formaldehyde and then stained for 15 min. The cells were then washed with PBS, and the slides were analyzed using a Nikon T*i* epi-fluorescence inverted microscope (Nikon Instruments Inc., NY, USA).

### 4.10. DNA Damage Assessment

HeLa cells (3 × 10^5^/mL) were incubated with emodin, vinblastine, and emodin and vinblastine for 24 h. Then the cells were trypsinized, fixed, and permeabilized with the reagents dedicated to the assay, then stained with anti-phospho-Histone H2A.X (Ser139) and anti-phospho-ATM (Ser1981) according to the instructions for the kit (Merck-Millipore, Guyancourt, France). Using the analyzer software module, the results were calculated and plotted as dot plots: percent negative cells (no DNA damage), percent cells with ATM activated, percent cells with H2AX activated, and percent DNA double breaks (dual activation both ATM and H2A.X). The experiment was repeated 3 times.

### 4.11. Assessment of Ultrastructural Changes

Cells at a density of 3 × 10^5^/mL for transmission electron microscopy were fixed with 3% glutaraldehyde in cacodyl buffer (Serva Electrophoresis GmbH, Heidelberg, Germany). Secondary fixation was performed in 2% osmium tetroxide (Spi, West Chester, PA, USA), after which the cells were dehydrated in alcohol in the concentration range of 10–99.8% and embedded in Epon 812 epoxy resin (Serva Electrophoresis GmbH, Germany). Polymerization was performed at 40 °C and 60 °C. The ultra-thin sections were cut on a Leica EM UC7 ultramicrotome (Leica Biosystems, Wetzlar, Germany) and additionally contrasted with uranium acetate and lead citrate. The preparations were analyzed with a Tecnai G2 Spirit transmission electron microscope (FEI Company, Hillsboro, OR, USA) using a Morada camera (Olympus, Soft Imagine Solutions, Münster, Germany). Mitochondria were measured using TEM Imaging & Analysis 3.2 SP6 software (FEI Company, Hillsboro, OR, USA). The size of organelles was assessed by measuring 100 cells each treated with the test compounds. Mean values were calculated from the results.

### 4.12. Assessment of Morphological Changes

Cells (controls and tested) at density of 1 × 10^5^/mL were grown on sterile coverslips in the dishes (Falcon). After exposure to the tested agents, the cells were fixed in methanol, stained with Harris hematoxylin and eosin, dehydrated using increasing alcohol series, and immersed in xylene. Morphological analysis was performed using a Nikon Eclipse 80i microscope with Nikon NIS Elements D 3.10 software (Nikon Instruments, Inc., Melville, NY, USA). The preparations were analyzed, counting 3000 cells in three independent experiments (9000 cells/concentration). The mitotic catastrophe was determined on the basis of indicators such as: giant cells, multinucleated cells, cells with micronuclei and abnormal mitotic figures.

### 4.13. Cell Cycle Analysis

Cells at a density of 3 × 10^5^/mL incubated for 24 h with emodin and vinblastine were fixed in ice-cold 70% ethanol, and then the cell cycle assay (Merck-Millipore, Guyancourt, France) was used. Following the staining procedure, the cells were analyzed using a Muse analyzer (Merck-Millipore, Guyancourt, France), determining the percentage of cells in each phase of the cycle. The experiment was repeated 3 times.

### 4.14. Fluorescent Labeling of F-Actin

To label the cytoskeleton components, cells were grown on sterile microscope slides in the culture dishes (Falcon, Fisher Scientific, Waltham, MA, USA). After 24 h of incubation, cells (1 × 10^5^/mL) were treated with emodin, vinblastine, and emodin and vinblastine for another 24 h and then fixed in 3.7% paraformaldehyde. Cells were washed in PBS and then permeabilized in 0.2% Triton X-100 (Sigma-Aldrich, St. Louis, MO, USA). Cells were then stained in the dark with fluorescein isocyanate-conjugated phalloidin (FITC) (Sigma Aldrich, St. Louis, MO, USA). Additionally, cell nuclei were stained with 4′,6-diamidin-2-phenylindole (DAPI) (Sigma Aldrich, St. Louis, MO, USA). Actin filaments were examined using a Nikon Eclipse T*i* inverted microscope (Nikon Instruments Inc., Melville, NY, USA) equipped with the Nikon A1 confocal laser system (Nikon Instruments Inc., Melville, NY, USA).

### 4.15. Statistical Analysis

The analysis of the obtained results was performed using one-way analysis of variance (ANOVA), with multiple post hoc comparisons using Tukey’s test. *p* < 0.05 was considered statistically significant. The Statistica 10.0 software (StatSoft, Krakow, Poland) was used for data analysis.

## 5. Conclusions

In summary, both emodin and vinblastine have shown cytotoxic effects on cervical cancer cells. These compounds lead to mitotic catastrophe and apoptosis, processes that are interrelated and very important in the elimination of cancer cells. Emodin in combination with other cytostatics may constitute a new form of anti-cancer therapy in the future, but this problem requires further research.

## Figures and Tables

**Figure 1 ijms-23-08510-f001:**
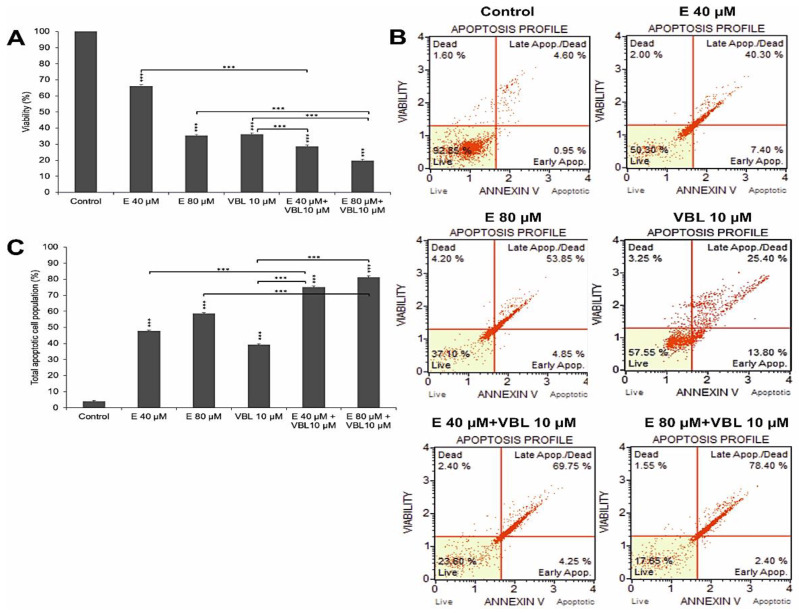
Emodin and vinblastine decrease viability and induce apoptosis in HeLa cells. Cells treated for 24 h with 40 and 80 µM concentrations of emodin, vinblastine (10 µM), and the combined action of 10 µM vinblastine with emodin at concentrations of 40 and 80 µM. (**A**) Effect of emodin and vinblastine on cell viability as assessed by MTT test. (**B**) Level of apoptosis assessed by annexin V-PE/7-AAD staining. Live cells (annexin V-PE-/7-AAD-), cells in early- (annexin V-PE+/7-AAD-) and late-stage apoptosis (annexin V-PE+/7-AAD+), and dead cells (annexin V-PE-/7-AAD+). (**C**) Percentage of apoptotic cells induced by tested agents. Differences statistically confirmed at: *** *p* < 0.001.

**Figure 2 ijms-23-08510-f002:**
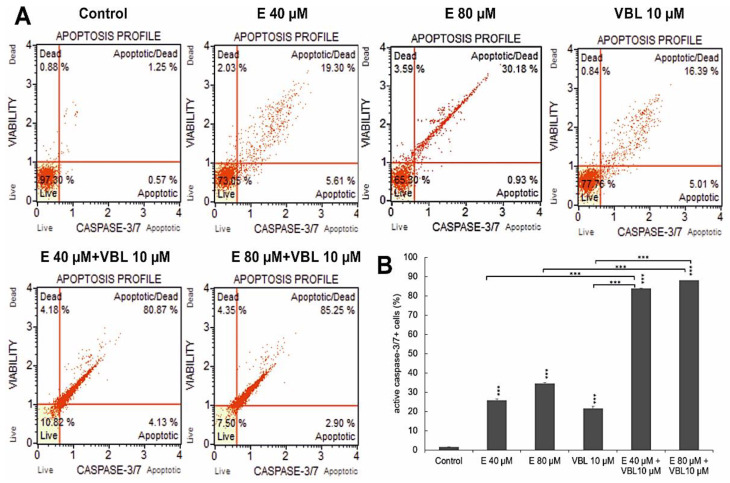
Changes in the caspase 3/7 activity induced by emodin and vinblastine. (**A**) Live cells (caspase 3/7-/7-AAD-), cells in early (caspase 3/7+/7-AAD-) and late apoptosis (caspase 3/7+/7-AAD+), dead cells (caspase 3/7-/7-AAD+). (**B**) Percentage of cells with active caspase 3/7. Differences statistically confirmed at: *** *p* < 0.001.

**Figure 3 ijms-23-08510-f003:**
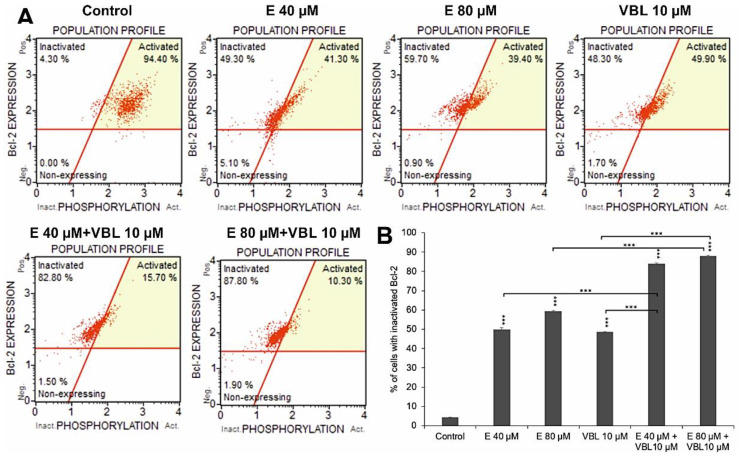
Dependence on the concentration of emodin and vinblastine percent of cells with inactivated Bcl-2 protein (**A**). Percentage of cells with Bcl-2 protein inactivation (**B**). The data in the graph show the mean ± SE of three independent experiments. Differences statistically confirmed at: *** *p* < 0.001.

**Figure 4 ijms-23-08510-f004:**
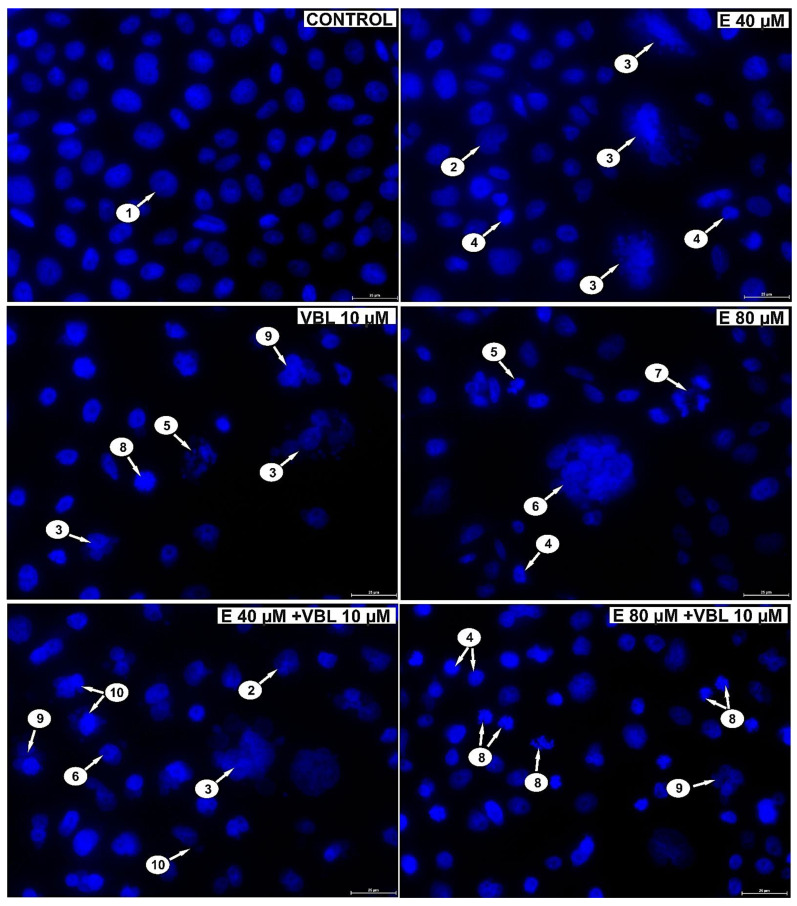
Emodin and vinblastine induce mitotic death. Evaluation of cell nucleus morphology using DAPI staining. HeLa cells incubated for 24 h with 40 and 80 µM emodin, 10 µM vinblastine and combined action of emodin and vinblastine (E 40 and 80 µM + 10 µM VBL). Explanation of markings: 1—interphase cells with normal nuclear morphology, 2—cells with altered cell nucleus shape, 3—multinucleated giant cells with micronuclei, 4—apoptotic cells with chromatin condensation, 5—apoptotic cells with nuclear fragmentation, 6—multinucleated giant cells, 7—abnormal mitotic figures (tripolar anaphase), 8—cells with abnormal chromosome segregation as an expression of cytoskeletal damage, 9—multinucleated cells, 10—cells with micronuclei. Magnification × 400.

**Figure 5 ijms-23-08510-f005:**
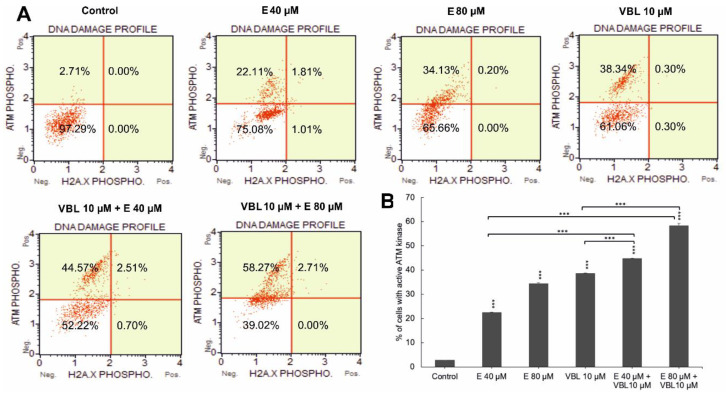
Degree of DNA damage induced by emodin and vinblastine. (**A**) HeLa cells treated for 24 h with emodin (40 and 80 µM), vinblastine (10 µM), and emodin and vinblastine (E 40 and 80 µM + 10 µM VBL) to induce DNA damage. Cells were stained with anti-phospho-histone H2A.X (Ser139) and anti-phospho-ATM (Ser1981) antibodies. (**B**) Percentage of cells with active ATM kinase. Data representative of three parallel experiments correspond to mean ± standard error (S.E.) values. Differences were statistically confirmed at: *** *p* < 0.001.

**Figure 6 ijms-23-08510-f006:**
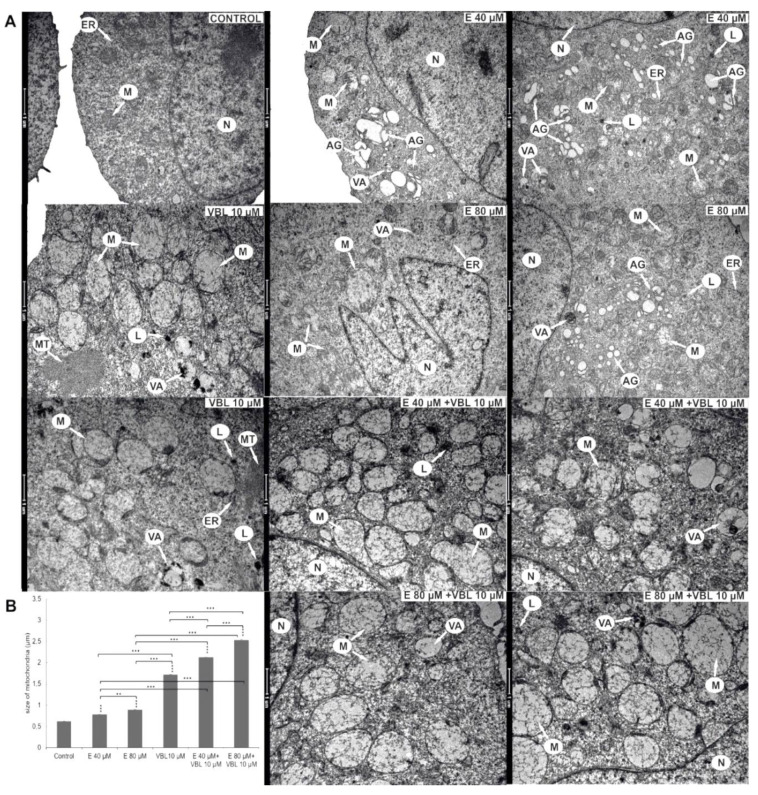
Ultrastructural changes in HeLa cells after 24 h of treatment with emodin, vinblastine and the combined action of both compounds. (**A**) Control—a cell with the correct nucleus and organelles with normal structure. Cells after the action of emodin (E 40 µM) with an increased number of Golgi apparatuses with swollen cisterns. Cells treated with emodin (E 80 µM)—extensive Golgi apparatuses with dispersed cisterns, numerous autophagic vacuoles, primary lysosomes and swollen mitochondria. Cells treated with vinblastine (VBL 10 µM)—the presence of swollen mitochondria with damaged cristae, autophagic vacuoles, lysosomes and cytoskeleton elements. Cells after the combined action of the compounds (E 40 µM and 80 µM + VBL 10 µM)—highly swollen and completely electron-transparent mitochondria with cristae drawn into the membrane, numerous lysosomes and autophagic vacuoles. Explanation of abbreviations: N—cell nucleus, M—mitochondria, VA—autophagic vacuoles, AG—Golgi apparatus, L—primary lysosomes, MT—microtubules, ER—rough endoplasmic reticulum. Magnification × 11,500. (**B**) Change in mitochondrial size as a function of the concentration of the test compounds. Data correspond to mean values ± standard error (S.E.) and are representative of three parallel experiments. The differences were statistically confirmed at: ** *p* < 0.01, *** *p* < 0.001.

**Figure 7 ijms-23-08510-f007:**
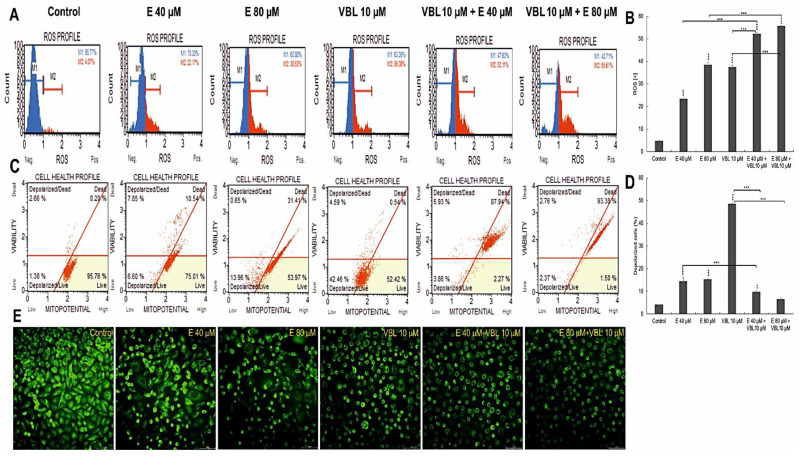
Assessment of ROS production and changes in mitochondrial potential in cells after treatment with vinblastine and emodin. (**A**) Generation of reactive oxygen species by 40 and 80 µM emodin, vinblastine (10 µM) and emodin in combination with vinblastine (E 40 µM + 10 µM VBL and E 80 µM+10 µM VBL). M1: negative cells, ROS (−); M2: cells with ROS activity, ROS (+). (**B**) Percentage of ROS (+) cells induced by test compounds. The degree of ROS production in HeLa cells was determined in comparison with the control. (**C**) Changes in mitochondrial membrane potential depending on the concentration of vinblastine and emodin. (**D**) Percentage of cells with mitochondrial membrane depolarization. Each sample was analyzed in triplicate. The differences were statistically confirmed at: ** *p* < 0.01, *** *p* < 0.001. (**E**) Analysis of changes in mitochondrial activity (rhodamine 123 staining). Visible control cells with high mitochondrial membrane potential and cells with reduced emission of fluorescent dye after the incubation with emodin, vinblastine and after combined action of compounds. Magnification × 400.

**Figure 8 ijms-23-08510-f008:**
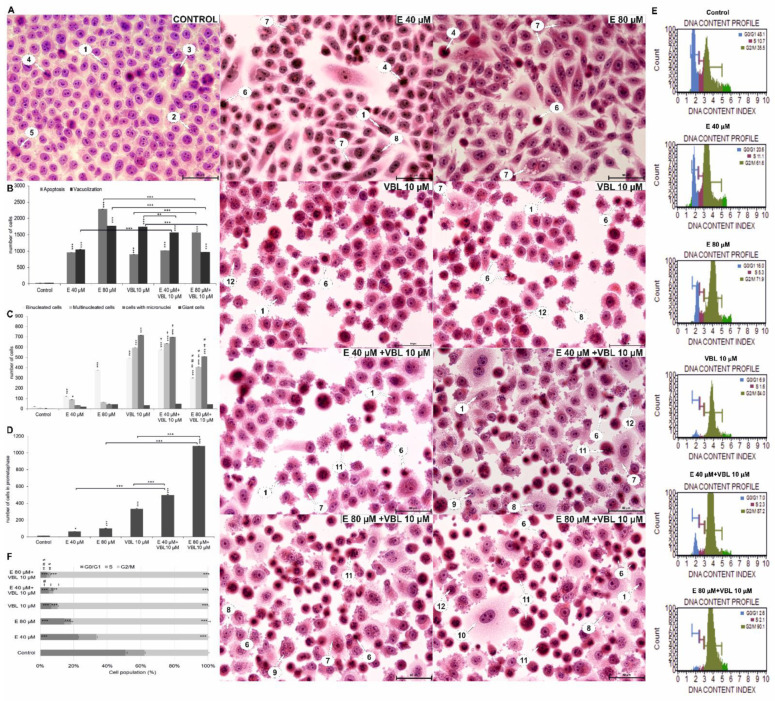
Morphological changes and changes in the cell cycle in Hela cells treated with emodin, vinblastine and their combined effects. (**A**) Cells stained with hematoxylin and eosin (H&E method). Control: cells in interphase and during cell division. Emodin 40 and 80 µM: numerous cells with cytoplasmic vacuolization, binuclear cells, multinucleated cells, and apoptotic cells. Vinblastine 10 µM: a large number of cells with micronuclei and cells with a disturbed formation of chromosomes. Combined action of emodin (40 and 80 µM) and vinblastine (10 µM): very numerous polynuclear cells, cells with micronuclei and cells with condensed chromosomes dispersed in the cytoplasm. 1—interphase; 2—prophase; 3—prometaphase; 4—metaphase; 5—telophase; 6—apoptosis; 7—vacuolization; 8—binuclear cells; 9—polynuclear cells; 10—giant cells; 11—cells with abnormal chromosome segregation; 12—cells with micronuclei. Magnification × 400. Comparison of the number of apoptotic cells and cells with cytoplasmic vacuolization (**B**) and indicators of mitotic catastrophe (**C**), i.e., the number of binucleated, multinucleated, giant cells and cells with micronuclei of HeLa line after 24 h of exposure to emodin (40 and 80 µM), vinblastine (10 µM), and their combined action (E 40 and 80 µM + 10 µM VBL). The symbol † represents statistically significant change (*p* < 0.0001) with emodin at a dose of 40 µM compared to cells treated by the combined action of emodin (40 µM) and vinblastine at a dose of 10 µM. The symbol ‡ represents statistically significant change (*p* ≤ 0.0001) with emodin at a dose of 80 µM compared to cells treated with the combined action of emodin (80 µM) and vinblastine at a dose of 10 µM. The symbol # represents statistically significant change (*p* < 0.05) with vinblastine in doses of 10 µM compared to cells treated with the combined action of emodin (40 µM) and vinblastine at a dose of 10 µM. The symbol ≠ represents statistically significant change (*p* < 0.0001) with vinblastine at a dose of 10 µM compared to cells treated with the combined action of emodin (80 µM) and vinblastine at a dose of 10 µM. (**D**) Number of cells with prometaphase as a function of the concentration of test compounds. (**E**) Emodin and vinblastine induce cell cycle arrest at the G2/M phase. Histograms showing cell distribution over the cycle after treatment with emodin and vinblastine. (**F**) Percentage of cells in the cell cycle as analyzed by flow cytometry. The symbol † represents statistically significant change (*p* < 0.0001) with emodin at a dose of 40 µM compared to cells treated with combined action of emodin (40 µM) and vinblastine at a dose of 10 µM. The symbol ‡ represents statistically significant change (*p* ≤ 0.0001) with emodin at a dose of 80 µM compared to cells treated with the combined action of emodin (80 µM) and vinblastine at a dose of 10 µM. The symbol # represents statistically significant change (*p* < 0.05) with vinblastine at a dose of 10 µM compared to cells treated with the combined action of emodin (40 µM) and vinblastine at a dose of 10 µM. The symbol ≠ represents statistically significant change (*p* < 0.0001) with vinblastine at a dose of 10 µM compared to cells treated with the combined action of emodin (80 µM) and vinblastine at a dose of 10 µM. The data in the graph shows the mean ± SE of three independent experiments. Differences statistically confirmed at: * *p* < 0.05; ** *p* < 0.01; *** *p* < 0.001.

**Figure 9 ijms-23-08510-f009:**
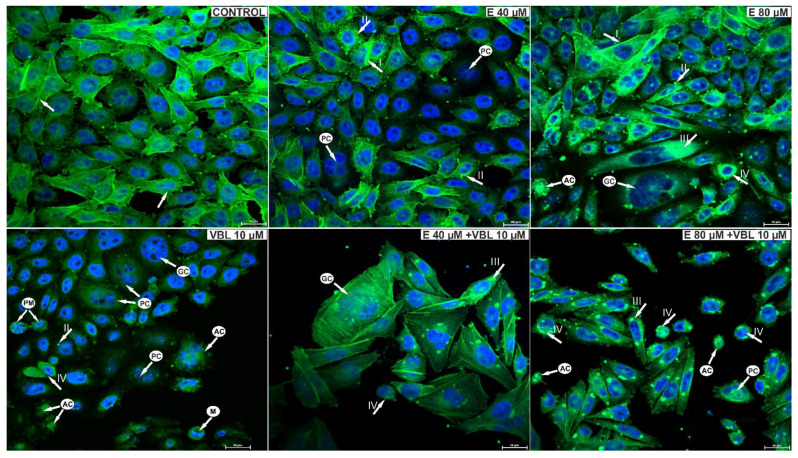
Visualization of cytoskeleton elements in cells labeled with FITC-conjugated phalloidin and cell nuclei using DAPI. Control cells, with regular distribution of actin filaments (arrow) and the correct shape of the cell nuclei. Cells treated with 40 μM emodin, with increased fluorescence of actin filaments, often marginally distributed (arrow I) and small deposits of actin in the cytoplasm (arrow II). Cells treated with 80 µM emodin with reorganization of the cytoskeleton and actin deposited in the cytoplasm (arrow III). Cells in prometaphase (arrow IV), rounded, with damaged cytoskeleton after treatment with vinblastine (10 µM). Cells after the combined action of vinblastine (10 µM) and emodin (40 µM and 80 µM) with visible damage to the cytoskeleton and the presence of actin deposits. Also visible are rounded cells with condensed actin, giant multinucleated cells with reorganization of the cytoskeleton as an expression of a mitotic catastrophe and apoptotic cells. Explanation of abbreviations: PM—prometaphase cells; M—cells in metaphase; PC—multinucleated cells; GC—giant cells; AC—apoptotic cells. Magnification × 400.

## Data Availability

The data that support the findings of this study are available from the corresponding author upon reasonable request.

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
