# Peer review of "Emodin Sensitizes Cervical Cancer Cells to Vinblastine by Inducing Apoptosis and Mitotic Death"

_ijms, 2022, doi:10.3390/ijms23158510_

Round 1

Reviewer 1 Report

Overall summary

The authors conducted a study to evaluate changes in cervical cancer cells after the combination of emodin with the chemotherapy drug vinblastine. According to the results, the treatment of HeLa cells with emodin potentiated the anticancer effect of vinblastine (lower cell viability, increased oxidative stress, higher mitotic catastrophe...).

General comments

The article is interesting, and it is quite well written. Furthermore, this manuscript could have great clinical, social and academic relevance. However, there are deficiencies which should be addressed before publication could be considered.

1.    The article should be reviewed for English language proficiency and grammar. There are sentences without sense (eg. lines 423-425…), typos (eg. lines 51, 105…). I think it would benefit from editing from a native English speaker.

2.    The authors must indicate that the effect of emodin has already been investigated on normal human cells and that it is not cytotoxic at concentrations used in this study. Likewise, the authors should explain why emodin was used at 40 and 80 µM.

3.     Materials and methods. Where appropriate, please include the cell densities that were used in the assays (eg. subsection 4.2. Assessment of cell viability).

4.    According to the figure 5C, the number of binucleated cells, multinucleated cells and cells with micronuclei was higher after the treatment with vinblastine than the combined treatment (vinblastine 10 µM and emodin 80 µM). Please, try to explain this in the discussion.

5.   References: The references list must be reviewed. For example, abbreviated journal names must be used (see https://www.mdpi.com/journal/ijms/instructions). On the other hand, numeration is duplicated and please, pay attention from the reference 59 onwards.

Other comments

1. Line 100. Change “annexin V-PE/PI staining” by “annexin V-PE/7-AAD staining”

2. Lines 127-128. The percentages 28.49 and 19.64% don´t represent dead cells, but live cells (see Figure 1F).

3.  All Figures must be numbered following their number of appearance. In the manuscript, Figure 5 (line 147) was citated before Figures 3 (line 215) and 4 (line 245).

4. Lines 299-300. You can delete the sentence “the data… <0.001”. The same one is at the end of the legend.

5. From the moment you clarify an abbreviation in the main text, you should use it (eg. ROS (line 437)).

Reviewer 2 Report

I recommend reconsidering of Manuscript “Emodin sensitizes cervical cancer cells to vinblastine through inducing apoptosis and mitotic death”  (ijms-1815373) for publishing after major revision, although journals’ recommendation for this iscontrol missing in some experiments”. My decision results first of all from 2 major reasons:

1.      The scope of this special issue is among others “This Special Issue aims to improve our understanding of the molecular mechanisms involved in apoptosis.” – the effect of emodin and vinblastine on cervical cancer cells including HeLa cells are well known, even Authors of the reviewed Manuscript have published such results, e.g. Anticancer Research February 2019, 39 (2) 679-686; DOI: https://doi.org/10.21873/anticanres.13163. So, this data are not novel. Maybe there is other Special Issue in IJMS or other MDPI journal which will fit better to this manuscript, i.e. is focused on studies on new anticancer drugs/therapies.

2.      Moreover, if activity of tested compounds is established, next step should be the establishment of the mechanisms which lead to the observed effects via functional studies i.e. down- or up-regulation of molecules which are probably involved in activity of tested compounds. Without these type of studies, it should not be concluded that emodin acts through inducing apoptosis and mitotic death but it could be only stated that emodin activates/leads to apoptosis and mitotic death.

3.      If the purpose of studies is to prove that the better outcomes are achieved by combined treatments, the results should be statistically analyzed not only in comparison with untreated control but also in comparison with cells treated separately with each compound and/or synergy/combination index should be calculated.

However, there are also some minor shortcomings:

1.      There is no explanation or any studies which indicate why both compounds were used at indicated concentrations.

2.      I recommend changing “dead cells” to “necrotic cells” in Figure 1 and in results description because apoptotic cells are also dead cells, whereas in this case this phrase is addressed only to necrotic cells without active caspase3/7. I also recommend describing the results presented in Figure 1 in one paragraph or prepare two separate figures for each paragraph.

3.      “in vitro “ and “in vivo” should be written in italics.

4.      I recommend presenting the results of MTT assay (cell viability, 2.4. paragraph) before apoptosis and caspase 3/7 assays because MTT assay is more general then apoptosis/caspase 3/7 assay considering the purpose of the assay.

5.      In my opinion it is hard to understand why photos of morphological changes observed after H&E staining are presented in Figure 5, whereas they are described together with results presented in Figure 2.

6.      I recommend unifying the expression used to indicate the observed changes, i.e. %, fold of change or number of cells.

7.      Caption of Figure 4 – indicate what does mean gates M1 and M2.

8.      Figure 5 E – describe gates with particular cell cycle phases.

9.      I recommend avoiding of figure indication in Discussion section.

10.  The role of DSBs and ATM kinase in mitotic death was discussed but not H2A.X although it was also assayed.

Round 2

Reviewer 2 Report

Dear Authors,

thank you very much for improving the manuscript according to my comments. However, in my opinion there are still some failures which need improvement. I recommend accepting after minor revision.

Previously my comment was: “If the purpose of studies is to prove that the better outcomes are achieved by combined treatments, the results should be statistically analyzed not only in comparison with untreated control but also in comparison with cells treated separately with each compound and/or synergy/combination index should be calculated.”

Of course Authors improve the analysis in relation with my comment, however it is made exaggeratedly. It is known that each compound of the tested compounds, emodin and vinblastine, present anticancer activities. But their combination hasn’t been evaluated yet. In the relation to the statement included in the title of manuscript, i.e. ”Emodin sensitizes cervical cancer cells to vinblastine through inducing apoptosis and mitotic death”, the purpose of the reviewed studies was to establish whether Emodin potentiates/sensitizes the activity of used, known cytostatic, vinblastine (combined treatment lead to better outcomes in comparison with single treatments). Hence, in my opinion, it is unnecessary presenting the comparison of e.g. E 40uM vs E 80uM, E40uM vs E80uM+VBL or E vs VBL. It is enough and reasonable to compare all samples with untreated control (as it was done) but also only the following samples:

-          40uM and VBL vs 40uM+VBL

-          80uM and VBL vs 80uM+VBL

to indicate the effect of E on activity of VBL. I recommend removing remaining statistical analysis from figures 1 – 5.

I’m still convinced that following paragraphs: 2.1. Emodin and vinblastine decrease cell viability; 2.2. Emodin and vinblastine induce cell apoptosis by activating caspase 3/7; 2.3. Emodin and vinblastine inactivate the Bcl-2 protein; 2.4. Emodin and vinblastine induce mitotic death; 2.5. Emodin and vinblastine induce phosphorylation of ATM kinase should correspond with separate figure. Hence, I recommend splitting Figure 1 and Figure 2 into single figures, corresponding with results described in each paragraph.

Best regards

Author Response

  1. According to the Reviewer's remark, the statistical analysis of the results has been corrected.
  2. Figures have been modified, ie Figures 1 and 2 have been divided into individual figures corresponding to the results described in each paragraph.

Thank you for Your very valuable comments that helped to improve our manuscript.

Kind regards